# Mechanism of Cell Death by Combined Treatment with an xCT Inhibitor and Paclitaxel: An Alternative Therapeutic Strategy for Patients with Ovarian Clear Cell Carcinoma

**DOI:** 10.3390/ijms241411781

**Published:** 2023-07-22

**Authors:** Urara Idei, Tsuyoshi Ohta, Hizuru Yamatani, Manabu Seino, Satoru Nagase

**Affiliations:** Department of Obstetrics and Gynecology, Faculty of Medicine, Yamagata University, Yamagata 990-9585, Japan; urara-ando@med.id.yamagata-u.ac.jp (U.I.); hizuru@med.id.yamagata-u.ac.jp (H.Y.); m-seino@med.id.yamagata-u.ac.jp (M.S.); nagases@med.id.yamagata-u.ac.jp (S.N.)

**Keywords:** ovarian clear cell carcinoma, drug resistant, glutamine/cystine transporter, paclitaxel, sulfasalazine, apoptosis, ferroptosis, targeted therapy

## Abstract

Ovarian clear cell carcinoma (OCCC) is a rare subtype of epithelial ovarian carcinoma that responds poorly to chemotherapy. Glutathione (GSH) is a primary antioxidant, which protects cells against reactive oxygen species (ROS). High levels of GSH are related to chemotherapeutic resistance. The glutamine/cystine transporter xCT is essential for intracellular GSH synthesis. However, whether xCT inhibition can overcome the resistance to chemotherapeutic agents in OCCC remains unclear. This study demonstrated that combined treatment with paclitaxel (PTX) and the xCT inhibitor sulfasalazine (SAS) significantly enhanced cytotoxicity more than the individual drugs did in OCCC cells. Treatment with PTX and SAS induced apoptosis more effectively than did individual drug treatments in the cells with significant generation of ROS. Moreover, combined treatment with PTX and SAS induced ferroptosis in the cells with low expression of glutathione peroxidase (GPx4), high levels of intracellular iron and significant lipid ROS accumulation. Therefore, our findings provide valuable information that the xCT inhibitor might be a promising therapeutic target for drug-resistant OCCC. The strategy of combined administration of PTX and SAS can potentially be used to treat OCCC and help to develop novel therapeutic methods.

## 1. Introduction

Ovarian clear cell carcinoma (OCCC) is a rare subtype of epithelial ovarian carcinoma (EOC) in Western countries; however, it accounts for more than 20% of EOC cases in Japan [1,2]. OCCC shows a poor stage-adjusted prognosis compared to other EOC subtypes and relatively high resistance to chemotherapy in advanced and recurrent cases [3,4]. Although the standard chemotherapy for EOC is platinum/taxane, the response rate to chemotherapy in OCCC is significantly lower than that in serous carcinoma [5]. Therefore, novel interventions beyond conventional chemotherapy are necessary to improve patient outcomes in advanced and recurrent OCCC cases.

OCCC harbors a high frequency of loss-of-function mutations in the AT-rich interactive domain 1A (ARID1A) [6,7], which encode a component of the SWI/SNF chromatin-remodeling complex. ARID1A-deficient cancer cells are specifically sensitive to the inhibition of glutathione (GSH) [8], which protects cells against reactive oxygen species (ROS)-induced cell death. Previously, we demonstrated using a metabolomics analysis that high levels of GSH are associated with resistance to paclitaxel in uterine serous carcinoma [9]. GSH synthesis requires cystine uptake, which involves the cystine–glutamate exchange transporter (xCT), and cystine is converted into cysteine within cells. Cysteine is the rate-limiting precursor substrate of GSH synthesis, and xCT controls intracellular cysteine levels [10]. xCT inhibition has been reported to overcome GSH-mediated resistance to chemotherapeutic agents in head and neck squamous carcinoma, lung cancer and colorectal cancer [11,12,13]. However, it remains unclear whether xCT inhibition can overcome resistance to chemotherapeutic agents in OCCC.

Sulfasalazine (SAS), an anti-inflammatory drug, is clinically used to treat bowel dis-ease and rheumatoid arthritis [14], and it exhibits anti-cancerous potential by inhibiting xCT [15]. SAS-induced GSH depletion leads to apoptosis and enhances sensitivity to cis-platin in colorectal and uterine cancer [13,16]. SAS-induced inhibition of xCT sensitizes cancer cells to ferroptosis in glioma and head and neck cancer [17,18]. Ferroptosis is a novel form of regulated cell death via the accumulation of lipid peroxidation products and ROS derived from iron metabolism, and it can be pharmacologically inhibited by iron chelators [19,20]. The key molecules of ferroptosis include xCT and glutathione peroxidase (GPX4), which effectively suppress lipid peroxidation [21,22]. We have previously shown that SAS overcomes resistance to paclitaxel through ferroptosis in uterine cancer [23]. Based on these previous findings, we hypothesized that SAS-induced inhibition of xCT may sensitize cancer cells to apoptosis in combination with cisplatin or to ferroptosis in combination with paclitaxel.

In the present study, we investigated whether SAS enhances the sensitivity of OCCC to cisplatin or paclitaxel. Furthermore, the molecular mechanisms of SAS-induced death of OCCC cells were elucidated.

## 2. Results

### 2.1. Effect of SAS on GSH Levels and ROS Accumulation in OCCC Cells

As a preliminary experiment, we calculated the IC50 of cisplatin (CDDP), PTX, and SAS in each cell line and set the concentrations below the IC50 in all the cell lines. The intracellular levels of GSH and ROS were measured to assess the inhibitory effects of SAS on xCT in SAS-treated OCCC cells. Treatment with 400 μM SAS for 24 h significantly decreased the intracellular GSH levels in all the OCCC cell lines (Figure 1A). Treatment with 400 μM SAS for 24 h increased the intracellular ROS levels in the TOV21G, RMG-1 and ES-2 cells, whereas no change was observed in the HAC-2 cells (Figure 1B).

### 2.2. Effect of SAS on Cytotoxicity of Cisplatin or Paclitaxel in OCCC Cells

To evaluate the effect of SAS on the cytotoxicity of CDDP or PTX, we examined the viability of OCCC cells treated with SAS, CDDP, or PTX for 24 h. Combined treatment with 10 µM CDDP and 400 µM SAS significantly decreased the viability of the ES-2 cells more than that when using their individual treatments (Figure 2A). Combined treatment with 10 nM PTX and 400 µM SAS significantly decreased the viability of the TOV21G, RMG-1 and ES-2 cells more than their individual treatments; however, such effects were not observed in the HAC-2 cells (Figure 2B). These results indicate that the combination of PTX and SAS induced cytotoxicity more effectively than the combination of cisplatin and SAS in OCCC cells, excluding the HAC-2 cells.

### 2.3. ROS Generation in OCCC Cells Treated with SAS and/or PTX

We examined the ROS levels in OCCC cells treated with PTX and/or SAS (Figure 3). Combined treatment with PTX and SAS increased the ROS generation more than their individual treatments in the TOV21G, RMG-1, and ES-2 cells. Combined treatment with PTX and SAS increased the ROS levels by approximately 5.2-, ~4.2-, and ~1.7-fold in the TOV21G, RMG-1 and ES-2 cells, respectively, compared with those in the vehicle-treated control cells. PTX and SAS significantly increased the ROS generation more than their individual treatments in the TOV21G and RMG-1 cells. However, the combination treatment of PTX with SAS failed to increase the ROS levels in the HAC-2 cells compared to those in the cells treated with vehicle, PTX, or SAS.

### 2.4. Apoptosis or Ferroptosis Underlying Cell Death Induced by Combined Treatment with PTX and SAS in OCCC Cells

To reveal the mechanism of cell death induced by combined treatment with PTX and SAS, we analyzed the expression of cleaved PARP, an apoptotic marker, by Western blot analysis and noticed that the combined treatment of PTX with SAS increased the levels of cleaved PARP in the TOV21G and RMG-1 cells, but not in the ES-2 and HAC-2 cells, compared with those in the cells treated with the vehicle, PTX or SAS individually (Figure 4A). We next investigated whether the ferroptosis inhibitor, ferrostatin-1, reverses the effects of SAS in increasing PTX cytotoxicity in OCCC cells (Figure 4B). OCCC cells were treated with or without PTX and SAS for 24 h in the absence or presence of ferrostatin-1. The combined treatment of ferrostatin-1 with PTX and SAS significantly increased the viability of the ES-2 cells compared to the PTX and SAS treatment. Ferrostain-1 did not reverse the cytotoxicity induced by treatment with PTX and SAS in other cells. To further confirm that the ES-2 cell death induced by the combination of PTX and SAS occurred through ferroptosis and not through apoptosis, cells were co-treated with PTX, SAS and the apoptotic inhibitor, Z-VAD-FMK. Z-VAD-FMK did not inhibit the effects of the PTX and SAS treatment on cell viability (Figure 4C). Therefore, the combined treatment of PTX and SAS enhanced the intracellular ROS accumulation, and it induced apoptosis in the TOV21G and RMG-1 cells with high levels of ROS accumulation and ferroptosis in the ES-2 cells with low levels of that. No changes in these parameters were observed in the HAC-2 cells.

### 2.5. The Effect of SAS Depends on the GSH Synthesis Pathway-Related Proteins in OCCC Cells and the Association between the Intercellular Iron Concentration and Ferroptosis Induced by Combined Treatment with PTX and SAS

To elucidate the differences in the effect of SAS on ROS accumulation and death in OCCC cells, the expression of proteins related to GSH synthesis was evaluated (Figure 5A). In this analysis, SKOV3, a serous carcinoma cell line, and Caov-3, a mucinous carcinoma cell line, were added as additional tissue subtypes of ovarian cancer. Western blot analysis showed similar effects of SAS on the xCT expression in OCCC and other cell subtypes. The levels of the glutamate-cysteine ligase catalytic subunit (GCLC), the rate-limiting enzyme of GSH synthesis, were similar in OCCC cells. The levels of cystathionine gamma-lyase (CGL) were higher in the HAC-2 cells than that in the other cell lines. We investigated the effect of the CGL inhibitor propargylglycine (PAG) on cytotoxicity induced by treatment with PTX and SAS in the HAC-2 cells. PAG significantly increased the growth inhibition induced by treatment with PTX and SAS in the HAC-2 cells (Figure 5B).

Combined treatment with PTX and SAS induced ferroptosis in the ES-2 cells only. The levels of GPx4 were low in the ES-2 cells (Figure 5A). We next investigated whether the combination of PTX and SAS induced ferroptosis in Caov-3, which expressed low GPX4 as well as ES-2 cells (Figure 5A). The combined treatment of ferrostatin-1 with PTX and SAS had no significant effect on viability compared with that exerted by the PTX and SAS treatment of the Caov-3 cells (Figure 5C). We observed that the intracellular Fe^2+^ levels in the Caov-3 cells were significantly lower than those in the ES-2 cells (Figure 5D). Therefore, we investigated whether ferroptosis was induced by combined treatment with PTX and SAS under an iron-loaded condition in the Caov-3 cells. With Fe^2+^ addition, the combined treatment of ferrostatin-1 with PTX and SAS significantly increased the viability compared to that exerted by the PTX and SAS treatment in the Caov-3 cells (Figure 5E). We next compared the level of iron among the OCCC cell lines. The intracellular Fe^2+^ levels in the ES-2 cells, which were sensitive to ferroptosis, were significantly lower than those in the HAC-2 cells and had no significant difference compared to those in the TOV21G and RMG-1 cells (Figure 5F). We also quantified the lipid ROS using BODIPY C11 in the ES-2 cells treated with PTX and/or SAS (Figure 5G). Treatment with SAS or PTX and SAS significantly increased the lipid ROS levels compared to those in the cells treated with the vehicle and PTX. These findings suggest that the induction of ferroptosis by combined treatment with PTX and SAS may be due to the low level of GPx4, high levels of iron, and lipid ROS accumulation in ovarian cancer cells.

### 2.6. The Effect of Treatment with PTX and SAS on Tumor Growth Inhibition

To confirm whether SAS enhances the antitumor efficacy of PTX in xenograft models of human OCCC cells, we administered PTX and/or SAS to nude mice subcutaneously inoculated with RMG-1 cells. The mice were randomly divided into four groups after the tumor’s length reached 10 mm. One group of mice (n = 4) was orally administered DMSO (control), the second group (n = 4) was intraperitoneally administered PTX (20 mg/kg), the third group (n = 5) was administrated SAS suspension (250 mg/kg), and the fourth group (n = 6) was administrated PTX and SAS. Five weeks after the drug administration, the tumor growth in the mice treated with the combination of PTX and SAS was significantly suppressed compared to that in the mice treated with DMSO, PTX, or SAS (Figure 6A). No side effects, such as weight loss, rash or diarrhea, were noticed (Figure 6B). We assessed the expression of caspase-3, a key mediator of apoptosis, by means of immunofluorescent staining in tumors treated with PTX and/or SAS and observed apoptosis in tumors treated with PTX and SAS (Figure 6C). Apoptosis was quantitated as a percentage of the cleaved caspase-3 positive cells compared to the total number of cells, and the combined treatment of PTX with SAS increased the immunofluorescent reactivity for cleaved caspase-3 compared to their individual treatments (Figure 6D).

## 3. Discussion

This is the first report to show that SAS could be a novel drug for enhancing cytotoxicity and cell death though co-administration with PTX in OCCC cells that are resistance to chemotherapeutic agents. The present study demonstrated that SAS enhanced the efficacy of PTX, but not CDDP, in inducing cytotoxicity in OCCC cells. CDDP generates intracellular ROS via a DNA-damaging effect and induces apoptosis in cancer cells [24]. PTX induces apoptosis by binding to β-mitotic tubulin to stabilize the microtubule organization and inhibiting mitosis and arresting it in metaphase [25]. SAS reduces intracellular GSH by inhibiting xCT and enhances ROS accumulation [15]. Considering the mechanism of action of each drug, we predicted that SAS might enhance the efficacy of CDDP rather than PTX. However, SAS enhanced the CDDP cytotoxicity in only one of four OCCC cell lines and enhanced the cytotoxicity of PTX in three of four cell lines. It remains unclear whether SAS is more cytotoxic in combination with PTX than CDDP in OCCC cell lines. Further study is needed to investigate the effect of combined treatment with SAS and CDDP in OCCC cells.

Treatment with PTX and SAS did not enhance cytotoxicity, ROS generation, or cell death in the HAC-2 cells compared to treatment with PTX and SAS in the other OCCC cell lines; therefore, the expression of protein enzymes related to the GSH production pathway was assessed to elucidate the underlying mechanism. The levels of cystathionine gamma-lyase (CGL), which synthesizes cysteine from methionine via the trans-sulfuration pathway [26], were higher in the HAC-2 cells than in the other cell lines. The high levels of CGL suggested that the combination of PTX and SAS did not induce ROS generation or cell death in the HAC-2 cells. Intracellular cysteine levels are primally controlled by xCT; however, the trans-sulfuration pathways synthesize cysteine from methionine, which is the major pathway of cysteine synthesis in glioma cells [27]. Upregulation of the trans-sulfuration pathway to compensate for cysteine depletion by xCT inhibitors reduces the efficacy of xCT inhibitor-induced cell death [28,29]. Although xCTs are expressed in most cancer cells [15], the expression of trans-sulfuration pathway-related enzymes varies among cancer types [30,31]. In this study, we observed similar xCT expression in four clear cell carcinoma cell lines; however, the expression of CGL was relatively higher in the HAC-2 cells. In the HAC-2 cells, inhibition of the trans-sulfuration pathway by PAGs enhanced the cytotoxicity of the combined administration of PTX and SAS. These findings suggest that SAS is less effective in HAC-2 cells, as their cysteine synthesis is dependent on the trans-sulfuration pathway, and that the effect of SAS can be predicted by CGL expression.

We explored the mechanisms of cell death induced by combined treatment with PTX and SAS. SAS has been reported to induce apoptosis [13,16] and ferroptosis [17,18] in cancer cells. The current study showed that combined treatment with PTX and SAS induced apoptosis in the TOV21G and RMG-1 cells and led to ferroptosis in the ES-2 cells. Although SAS induces GSH depletion, its effect on ROS generation has been reported to differ among cancer cell types [32,33]. Therefore, we hypothesized that the mechanism of cell death may differ depending on the levels of ROS induced by combined treatment with PTX and SAS. In this study, the combined treatment of PTX with SAS increased the ROS generation in the TOV21G, RMG-1 and ES-2 cells. The extent of the ROS generation was significantly higher in the cells with combined treatment-induced apoptosis (TOV21G and RMG-1), whereas it was low in the cells without induced apoptosis (ES-2 cells). The key molecules of ferroptosis include GPX4, which effectively suppresses lipid peroxidation [21,22]. Ferroptosis is dependent on intracellular iron [19]. GSH depletion is essential for xCT inhibitor-induced ferroptosis, and GPx4 activity and intracellular iron levels are associated with sensitivity to ferroptosis in cancer cells [19,20,21,22,34,35]. The levels of GPx4 were lower in the ES-2 cells than in the other OCCC cells, suggesting the enhanced sensitivity of ES-2 cells to GSH depletion and ferroptosis. The inhibition of ferroptosis did not affect the cytotoxicity of combined treatment with PTX and SAS in the Caov-3 cells with lower GPx4 expression. The intracellular Fe^2+^ level significantly decreased in the Caov-3 cells compared to the ES-2 cells; therefore, ferroptosis was induced by the combination of PTX and SAS under an iron-loaded condition in the Caov-3 cells. The carcinogenesis of OCCC is related to oxidative stress generated from the free iron of endometriosis [36], and our study showed that the intracellular iron in OCCC cells was higher than that in other subtypes of ovarian cancer cells. However, the intracellular iron levels in the ES-2 cells, which were sensitive to ferroptosis, were significantly lower than those in the HAC-2cells and had no significant difference compared to those in the TOV21G, and RMG-1 cells. These results suggest that a high level of intracellular iron was not the rate-limiting factor of SAS-induced ferroptosis in OCCC cells. The lipid ROS plays an important role in inducing ferroptosis [21,22], and treatment with SAS or SAS and PTX significantly increased the lipid ROS levels compared to those in the cells treated with the vehicle and PTX in the ES-2 cells. Taken together, combined treatment with PTX and SAS induces apoptosis via high levels of ROS generation, whereas it induces ferroptosis in the cells with low GPx4 expression and high levels of iron and lipid ROS accumulation, even if the ROS generation is low.

In clinical trials for cancer treatment, SAS alone was not effective in patients with advanced or recurrent malignant glioblastoma [37]; however, in patients with advanced non-small-cell lung cancer, the combination of SAS and CDDP or pemetrexed resulted in prolonged progression-free survival [38]. Therefore, SAS combined with existing anti-cancer drugs may improve the conditions of cancer patients. In the present study, SAS increased the cytotoxicity of PTX at a dose of 400 μM in OCCC cell lines. This dose is equivalent to an SAS dose of 11.2 g/day, and it is ~3.2-fold higher than that used in clinical treatment [39]. Thus, the development of more potent xCT inhibitors than SAS is required for the clinical treatment of patients with recurrent OCCC.

## 4. Materials and Methods

### 4.1. Cell Cultures

The RMG-1 cells were provided by Dr. Nozawa at the Department of Gynecology and Obstetrics, Keio University Graduate School of Medicine (Tokyo, Japan). The ES-2 cells were provided by Dr. Yaegashi at the Department of Gynecology and Obstetrics, Tohoku University Graduate School of Medicine (Sendai, Japan). The HAC-2 cells were provided by Dr. Nishida at the Department of Gynecology and Obstetrics, Tsukuba University Graduate School of Medicine (Ibaraki, Japan). The TOV21G, Caov-3 and SKOV-3 cells were obtained from the American Type Culture Collection (Manassas, VA, USA). The RMG-1, ES-2, HAC-2, and TOV21G cell lines are all clear cell carcinoma cell lines. The SKOV3 cell line is a serous carcinoma cell line, and the Caov-3 cell line is a mucinous carcinoma cell line.

The RMG-1, HAC-2 and ES-2 cell lines were maintained in DMEM/F12 medium (Thermo Fisher Scientific, Inc.; Tokyo, Japan) supplemented with 10% fetal bovine serum (FBS; Sigma-Aldrich; Tokyo, Japan) and 1% penicillin-streptomycin (Sigma-Aldrich; Tokyo, Japan) at 37 °C in a humidified atmosphere with 5% CO_2_. The TOV21G and SKOV-3 cell lines were maintained in a medium composed of a 1:1 combination of MCDB105 (Sigma-Aldrich; Tokyo, Japan) and Medium 199 (Sigma-Aldrich; Tokyo, Japan) supplemented with 10% (for SKOV-3) or 15% (for TOV21G) FBS and 1% penicillin-streptomycin at 37 °C in a humidified atmosphere with 5% CO_2_. The Caov-3 cells were maintained in DMEM medium (Thermo Fisher Scientific, Inc.; Tokyo, Japan) supplemented with 10% FBS and 1% penicillin-streptomycin at 37 °C in a humidified atmosphere with 5% CO_2_.

### 4.2. Antibodies and Reagents

The following antibodies were purchased against: xCT (1:1000; cat. no. ab37185; Abcam; Cambridge, UK), cleaved-PARP (1:1000; cat. no. 9541; Cell Signaling Technology; Danvers, MA, USA), GPx4 (1:1000; cat. no. ab125066; Abcam; Cambridge, UK), cystathionine gamma-lyase (CGL; 1:1000; cat. no. 12217-1-AP; Proteintech; San Antonio, TX, USA); GCLC (1:1000; cat. no. GTX16315; Gene Tex, Inc.; San Antonio, TX, USA), β-actin (1:10,000; cat. no. A2228; Sigma-Aldrich; Tokyo, Japan), and cleaved caspase-3 (1:400; cat. no. 9661; Cell Signaling Technology; Danvers, MA, USA).

The following reagents were purchased and dissolved in DMSO to prepare the stock solutions: 100 mM SAS, 10 µM paclitaxel, 10 mM cisplatin, 10 mM 2′,7′-dichlorofluorescin diacetate (DCFH-DA), 1 M propargylglycine (PAG) and 10 mM ferrostatin-1 (all dissolved in DMSO and from Sigma-Aldrich; Tokyo, Japan). A total of 10 mM Z-VAD-FMK (Peptide Institute, Inc.; Osaka, Japan) was dissolved in DMSO. The BODIPY C11 [4,4-difluoro-5-(4-phenyl-1,3-butadienyl)-4-bora-3a,4a-diaza-s-indacene-3-undecanoic acid] was purchased from Thermo Fisher Scientific, Inc.

### 4.3. Cell Viability Assays

The cells were seeded into 96-well plates at 5 × 10^3^ cells per well and incubated at 37 °C for 24 h. They were then treated with SAS and other drugs and incubated at 37 °C for 24 h. The cell numbers were determined based on our previous study [16]. Cell viability was assessed using the tetrazolium compound MTS [3-(4,5-dimethylthiazol-2-yl)-5-(3-carboxymethoxyphenyl)-2-(4-sulfophenyl)-2H-tetrazolium, inner salt] (Promega; Madison, WI, USA). MTS assays were performed and then the samples were incubated with CellTiter 96^®^ AQueous One Solution Reagent (Promega; Madison, WI, USA) for 90 min at 37 °C. Subsequently, the 490 nm absorbance was measured using a Model 680 Microplate Reader^®^ (Bio-Rad Laboratories, Inc., Hercules, CA, USA). Cell viability was calculated from the ratio of cells treated with each of the drugs to that of untreated cells, which was set as 1 (mean ± SD; n = 8). Untreated cells were used as a control.

### 4.4. Glutathione Analysis

The GSH levels in cells exposed to various drugs for 24 h were measured using a GSH-GloTM Glutathione assay according to the manufacturer’s protocol (Promega; Madison, WI, USA). The cells were seeded into white 96-well plates at 5 × 10^3^ cells per well and incubated at 37 °C for 3 h. The luminescent signal was measured using a Thermo Scientific Varioskan^®^ Flash (Thermo Fisher Scientific Inc., Tokyo, Japan). The GSH levels were calculated from the ratio of cells treated with drugs to that of the untreated cells, which was set as 1 (mean ± SD; n = 8).

### 4.5. ROS and Lipid ROS Measurements

For measuring the ROS generation, the cells were exposed to 10 μM DCFH-DA (Sigma-Aldrich; Tokyo, Japan) for 10 min at room temperature. Care was taken to shield the cells from light during the procedures. Cells exhibiting a signal for DCFH-DA above the gate established using the isotype control treated without DCFH-DA were deemed ROS-positive. The cells were then used in the FACS analysis to quantify the intensity of the DCF fluorescence using a FACSCantoTM II Flow Cytometer (BD Biosciences, Franklin Lakes, NJ, USA), and the data were analyzed using FlowJo software, version 7.6.5 (Treestar Inc., Ashland, OR, USA).

For the lipid ROS measurement, the cells were exposed to 10 μM BODIPY C11 (Thermo Fisher Scientific, Inc.; Tokyo, Japan) for 15 min at 37 °C. The cells were protected from light during the respective procedures. Cells exhibiting a signal for BODIPY C11 above the gate established using the isotype control treated without BODIPY C11 were deemed lipid ROS-positive. These cells were subsequently used in the FACS analysis to quantify the intensity of the DCF fluorescence using a FACSCantoTM II flow cytometer. All the experiments were carried out in quadruplicate.

### 4.6. Immunoblotting Analysis

The cells were washed with cold PBS twice and then lysed in RIPA buffer (FUJIFILM Wako Pure Chemical Corporation, Osaka, Japan). The protein concentration was determined using a DCTM protein assay kit (Bio-Rad Laboratories, Inc., Hercules, CA, USA). Cell lysates containing equal amounts of protein were separated using 5% SDS-PAGE and transferred onto polyvinylidene difluoride membranes. Blocking was performed using 5% skimmed milk powder or 3% bovine serum albumin (Sigma-Aldrich; Tokyo, Japan) in 1× TBS at room temperature for 1 h. The membrane was sequentially probed with the aforementioned primary antibodies overnight at 4 °C, and then with an appropriate horseradish peroxidase (HRP)-conjugated secondary antibody at room temperature for 1 h, according to the manufacturer’s protocols. The secondary antibodies used were: Mouse IgG HRP Linked Whole Ab (1:5000; cat. no. NA931-1ML) and Rabbit IgG HRP Linked Whole Ab (1:5000; cat no. NA934-1ML), both purchased from Cytiva. The immunoreactive bands were visualized using ECL Prime Western Blotting Detection Reagent (GE Healthcare Life Sciences; Buckinghamshire, England).

### 4.7. Detection of Intracellular Fe^2+^

The intracellular Fe^2+^ were measured using FerroOrange (Dosindo, Kumamoto, Japan). The samples were seeded on a 96-well plate and cultured overnight in a 37 °C incubator equilibrated with 95% air and 5% CO_2_. The supernatant was discarded and the cells were washed with HBSS three times. Next, 1 μmol/L FerroOrange working solution was added to the cells and they were incubated in a 37 °C incubator equilibrated with 95% air and 5% CO_2_. Subsequently, the fluorescent intensities (Ex: 543 nm, Em: 580 nm) were detected using a Thermo Scientific Varioskan^®^ Flash (Thermo Fisher Scientific Inc., Tokyo, Japan).

### 4.8. Animal Experiments

The procedures involving animals used in the present study were approved by the Animal Care Committee of Yamagata University (approval no. 31164) in accordance with institutional and Japanese government guidelines for animal experiments. The RMG-1 cells were harvested in 0.25% trypsin/PBS/EDTA, washed once each with medium and PBS, and resuspended in PBS at 1 × 10^6^ cells/100 µL. Next, 2 × 10^6^ cells were injected subcutaneously into the flanks of 6-week-old female BALB/cAJcl-nu/nu mice (n = 24), and the mice were randomly divided into four groups. In the subcutaneous model, once the tumor’s length reached 10 mm (about 14–40 days after implantation, n = 19, 5 mice developed no tumor), the mice were administrated with each of the drugs. One group of mice (n = 6) was treated with paclitaxel (20 mg/kg BW) 3 times weekly plus SAS (200 mg/kg BW) 5 times weekly for 4 weeks. A second group of mice (n = 4) was treated with paclitaxel alone (20 mg/kg BW) 3 times weekly for 4 weeks. The third group of mice (n = 5) was treated with SAS (200 mg/kg BW) 5 times weekly for 4 weeks. The remainder of the mice (n = 4) received the vehicle (DMSO + PBS) alone.

### 4.9. Immunofluorescent Staining

The paraffin-embedded tissue sections were deparaffinized in xylene and then rehydrated using alcohol. Following deparaffinization, the sections were placed in trisodium citrate buffer (0.1 M trisodium citrate and 0.1 M trisodium citrate dehydrate) and heated twice for 15 min in a 1 kW microwave oven for antigen retrieval. Endogenous peroxidase activity was blocked via incubation for 15 min with 3% H2O2 solution in methanol (0.01 M) at room temperature. Non-specific binding was quenched via 30 min incubation in 3% skimmed milk powder in phosphate-buffered saline (PBS) at room temperature. After an additional wash with PBS, the sections were incubated with the primary antibodies overnight at 4 °C. Anti-cleaved caspase-3 antibodies were used to assess apoptosis. After washing, the sections were incubated with Alexa Fluor 488-labeled goat anti-rabbit antibody (1:1000; cat. no. 4412; Cell Signaling Technology; Danvers, MA, USA) for 1 h at room temperature, followed by staining with propidium iodide (1 μg/mL; Thermo Fisher Scientific, Inc.; Tokyo Japan) for 15 min at room temperature in the dark. The fluorescence was visualized using a fluorescence microscope (Olympus, Tokyo, Japan). For the apoptosis assays, four random fields per section were recorded at × 400 magnification, and the stained cells were counted. The number of cleaved caspase-3-positive cells was expressed as the percentage of total cells counted.

### 4.10. Statistical Analysis

The statistical analysis was performed using GraphPad Prism software (version 5.0; GraphPad Software, Inc.; Boston, MA, USA). Data are presented as the mean ± SD. An unpaired Student’s *t*-test was used to compare the differences between two groups, while a one-way followed by a Bonferroni post hoc test were used to compare the differences between multiple groups. A *p* < 0.05 was considered to indicate a statistically significant difference and represented by an asterisk or different letters in the figures.

## 5. Conclusions

In summary, the xCT inhibitor SAS enhanced the efficacy of PTX in OCCC cell lines. Combined treatment with PTX and SAS induced apoptosis through the generation of excessive reactive oxygen species (ROS) or ferroptosis through the low expression of glutathione peroxidase (GPx4) and high levels of intracellular iron and lipid ROS accumulation. However, high CGL levels compensated for SAS-induced GSH depletion and did not enhance the anti-tumor effect of PTX combined with SAS. Therefore, our findings provide valuable information that the xCT inhibitor might be a promising therapeutic target for drug-resistant OCCC. The combination of SAS and PTX may be a potential therapy for OCCC and should be further cultivated to develop novel therapeutic methods.

## Figures and Tables

**Figure 1 ijms-24-11781-f001:**
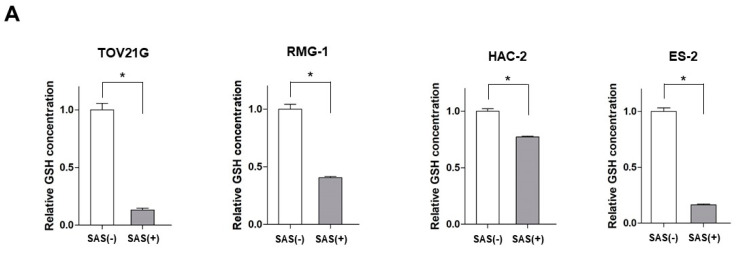
The effect of SAS on the intracellular GSH levels and ROS accumulation in OCCC cell lines. (**A**) GSH levels were compared with or without 400 μM SAS treatment for 24 h in OCCC cells. GSH levels were calculated from the ratio of cells treated with drugs to that of untreated cells set as 1 (mean ± SD; *n* = 8). Data are presented as the mean ± standard deviation (SD). * *p* < 0.05 vs. un-treated cells. (**B**) OCCC cells were treated with 400 μM SAS for 24 h and the intracellular ROS levels were analyzed by means of flow cytometry after staining with DCFH-DA. GSH, glutathione; ROS, reactive oxygen species; SAS, sulfasalazine; DCFH-DA, 2′,7′-dichlorofluorescin diacetate. −, without; +, with.

**Figure 2 ijms-24-11781-f002:**
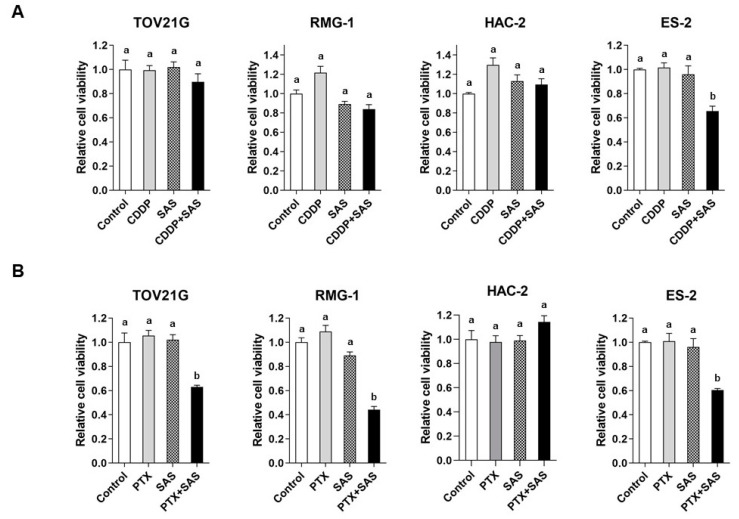
The effect of SAS combined with cisplatin or paclitaxel on the viability of OCCC cell lines. TOV21G, RMG-1, HAC-2 and ES-2 cells were treated with (**A**) 10 µM CDDP and/or 400 µM SAS, and (**B**) 10 nM paclitaxel and/or 400 µM SAS for 24 h, and the cell viability was assessed using an MTS assay. Cell viability was calculated from the ratio of the absorbance of cells treated to that of the untreated cells (set as 1) (n = 8). Data are presented as the mean ± SD. Different letters above the bars indicate a significant difference (*p* < 0.05, n = 8, in each group). SAS, sulfasalazine; CDDP, cisplatin; PTX, paclitaxel.

**Figure 3 ijms-24-11781-f003:**
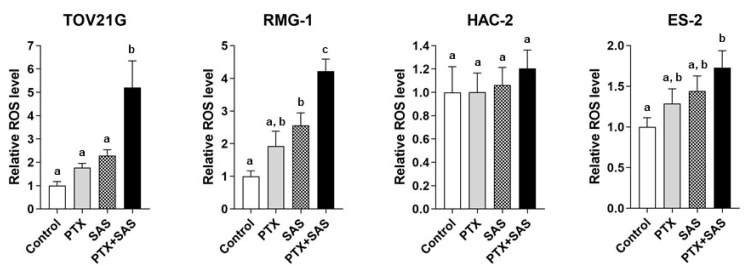
ROS generation by the combined treatment of SAS and PTX in OCCC cell lines. TOV21G, RMG-1, HAC-2 and ES-2 cells were cultured for 24 h in the presence of 10 nM PTX and/or 400 μM SAS and the intracellular ROS levels were analyzed by means of flow cytometry after staining with DCFH-DA. The ROS levels were calculated from the ratio of the intensity of the DCF fluorescence of treated cells to that of untreated cells (set as 1) (n = 5). Data are presented as the mean ± SD. Different letters above the bars indicate a significant difference (*p* < 0.05, n = 5, in each group). ROS, reactive oxygen species; PTX, paclitaxel; SAS, sulfasalazine; DCFH-DA, 2′,7′-dichlorofluorescin diacetate.

**Figure 4 ijms-24-11781-f004:**
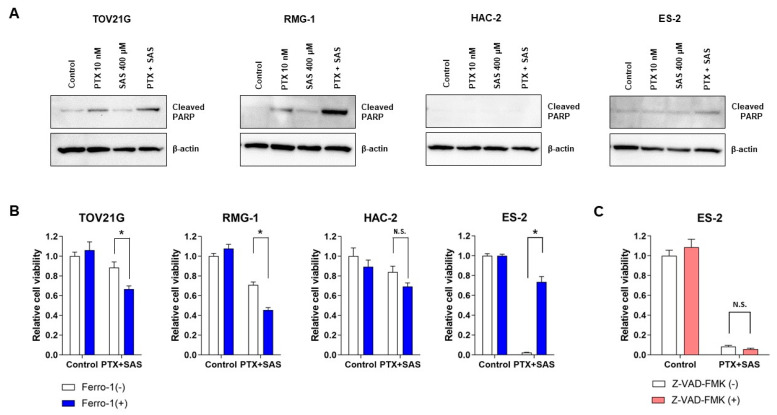
Assessment of cell death induced by the combined treatment of PTX and SAS in OCCC cell lines. (**A**) TOV21G, RMG-1, HAC-2 and ES-2 cells were treated with 10 nM PTX and/or 400 μM SAS for 24 h and the cell lysates were probed for anti-cleaved-PARP antibody using Western blot analysis. β-actin was used as the internal control. PTX, paclitaxel; SAS, sulfasalazine. (**B**) TOV21G, RMG-1, HAC-2 and ES-2 cells were treated with or without 10 nM PTX and 400 μM SAS for 24 h in the absence or presence of ferrostatin-1 (1 μM) and cell viability was assessed using an MTS assay. (**C**) ES-2 cells were treated with or without 10 nM PTX and 400 μM SAS for 24 h in the absence or presence of Z-VAD-FMK (20 μM) and cell viability was assessed using an MTS assay. Cell viability was calculated from the ratio of the absorbance of cells treated to that of the untreated cells (set as 1) (n = 8). Data are presented as the mean ± SD. * *p* < 0.05. SAS, sulfasalazine; PTX, paclitaxel; −, without; +, with. N.S.; not significant.

**Figure 5 ijms-24-11781-f005:**
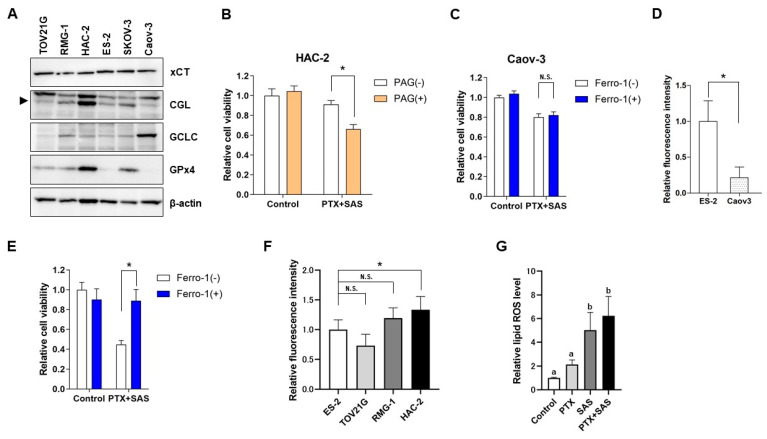
Expression of proteins related to the GSH synthesis pathway in OCCC cell lines and association between the intercellular iron concentration and ferroptosis induced by the combined treatment of PTX and SAS. (**A**) Levels of xCT, CGL, GCLC and GPx4 activation were examined in TOV21G, RMG-1, HAC-2, ES-2, SKOV-3 and Caov3 cells. Lysates were immunoblotted with anti-xCT, anti-CGL, anti-GCLC, anti-GPx4, and anti-β-actin antibodies; β-actin was used as an internal control. (**B**) Viability of HAC-2 cells treated with or without 10 nM PTX and 400 μM SAS for 24 h in the absence or presence of 1 mM PAG. Cell viability was calculated from the ratio of the absorbance of cells treated to that of the untreated cells (set as 1) (n = 8). Data are presented as the mean ± SD. * *p* < 0.05. (**C**) Viability of Caov-3 cells treated with or without 10 nM PTX and 400 μM SAS for 24 h in the absence or presence of ferrostatin-1 (1 μM). Subsequently, cell viability was assessed using an MTS assay. Cell viability was calculated from the ratio of the absorbance of cells treated to that of the untreated cells (set as 1) (n = 8). Data are presented as the mean ± SD. N.S; not significant. (**D**) Caov-3 and ES-2 cells were cultured for 24 h and the intracellular iron levels were calculated from the ratio of the absorbance of these cells to that of ES-2 cells (set as 1) (n = 6). * *p* < 0.05 vs. ES-2 cells. (**E**) Viability of Caov-3 cells cultured in the presence of 100 µM ammonium iron (II) sulfate hexahydrate for 30 min and subsequently treated with or without 10 nM PTX and 400 μM SAS for 24 h in the absence or presence of ferrostatin-1 (1 μM). Cell viability was calculated from the ratio of the absorbance of cells treated to that of the untreated cells (set as 1) (n = 8). Data are presented as the mean ± SD. * *p* < 0.05. (**F**) OCCC cell lines were cultured for 24 h and the intracellular iron levels were calculated from the ratio of the absorbance of these cells to that of ES-2 cells (set as 1) (n = 6). Data are presented as the mean ± SD. * *p* < 0.05 vs. ES-2 cells. N.S; not significant. (**G**) ES-2 cells were cultured for 24 h in the presence of 10 nM PTX and/or 400 μM SAS and the lipid ROS levels were analyzed by means of flow cytometry after staining with BODIPY C11. Lipid ROS levels were calculated from the ratio of the intensity of DCF fluorescence of treated cells to that of untreated cells (set as 1) (n = 5). Data are presented as the mean ± SD. Different letters above the bars indicate a significant difference (*p* < 0.05, n = 5, in each group). SAS, sulfasalazine; PTX, paclitaxel; PAG, propargylglycine; −, without; +, with.

**Figure 6 ijms-24-11781-f006:**
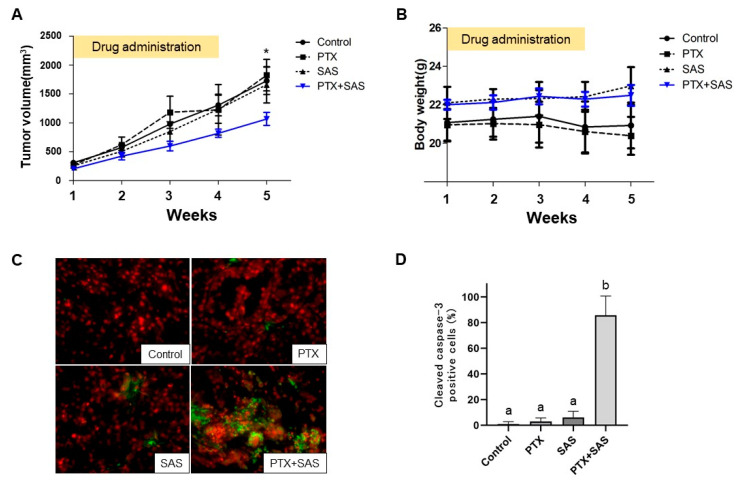
PTX and SAS suppress xenograft tumor growth. (**A**) Time course and volume of tumors formed by RMG-1 cells in BALB/cAJcl-nu/nu mice treated with the individual drugs. RMG-1 cells were injected subcutaneously into the flanks of 6-week-old female BALB/cAJcl-nu/nu mice (n = 24), and the mice were randomly divided into four groups. In the subcutaneous model, once the tumor’s length reached 10 mm (n = 19, 5 mice developed no tumors), the mice were administrated with each of the drugs. One group of mice (n = 6) was treated with paclitaxel (20 mg/kg BW) 3 times weekly plus SAS (200 mg/kg BW) 5 times weekly for 4 weeks. A second group of mice (n = 4) was treated with paclitaxel alone (20 mg/kg BW) 3 times weekly for 4 weeks. The third group of mice (n = 5) was treated with SAS (200 mg/kg BW) 5 times weekly for 4 weeks. The remainder of the mice (n = 4) received the vehicle (DMSO + PBS) alone and were defined as the control. Values shown represent means ± SD. * *p* < 0.05 vs. mice with combination treatment of PTX and SAS. (**B**) Changes in weights of the mice with time. Values shown represent means ± SD. (**C**) Paraffin-embedded tissue sections were deparaffinized in xylene and double-stained with anti-cleaved caspase 3 followed by FITC-conjugated goat anti-rabbit IgG (green) and PI staining (red). Magnification × 400. (**D**) The number of cleaved caspase-3 positive cells was expressed as the percentage of total cells and shown for each tumor treated with DMSO + PBS (control), PTX and/or SAS. Values shown represent means ± SD. Different letters above the bars indicate a significant difference (*p* < 0.05, n = 4, in each group). SAS, sulfasalazine; PTX, paclitaxel.

## Data Availability

The datasets used and/or analyzed during the current study are available from the corresponding author on reasonable request.

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
