# Peer review of "Mechanism of Cell Death by Combined Treatment with an xCT Inhibitor and Paclitaxel: An Alternative Therapeutic Strategy for Patients with Ovarian Clear Cell Carcinoma"

_ijms, 2023, doi:10.3390/ijms241411781_

Round 1

Reviewer 1 Report

Title: Mechanism of cell death by combined treatment of xCT inhibi- 2 tor and paclitaxel: An alternative therapeutic strategy for pa- 3 tients with ovarian clear cell carcinoma

This study of Urara Idei shows that xCT inhibitor is can be used for the inhibition of GSH synthesis at the ovarian clear cell carcinoma level, and consequently prevention of chemotherapeutic resistance or even the death of carcinogenic cells. The results obtained by this study are relevant and can be a strategy for improving the treatment of ovarian cancer. In ny ponion it is an important study, but requires some revision:

Abstract:

Please mention the effect of xCT inhibitor sulfasalazine on different parameters measured with the significance

Results:

Please mention the significance of the effect of SAS on the indices measured

Discussion:

According to your results, xCT inhibitor causes an accumulation of ROS at the cell level. According to subsequent studies, these ROS cause harmful effects on non-carcinogenic cells and therefore affect the functioning of organs? Also xCT inhibitor can attack normal non-carcinogenic cells and cause damage and mutations? Thank you for discussing these hypotheses

Author Response

Reviewer 1

This study of Urara Idei shows that xCT inhibitor is can be used for the inhibition of GSH synthesis at the ovarian clear cell carcinoma level, and consequently prevention of chemotherapeutic resistance or even the death of carcinogenic cells. The results obtained by this study are relevant and can be a strategy for improving the treatment of ovarian cancer. In my onion it is an important study, but requires some revision:

Thank you for your thoughtful review of our manuscript. Please find our detailed comments regarding your suggestions below.

Abstract:

Please mention the effect of xCT inhibitor sulfasalazine on different parameters measured with the significance

As you suggested, we described it in lines 14-19.

Results:

Please mention the significance of the effect of SAS on the indices measured

We described it in lines 110-111.

Discussion:

According to your results, xCT inhibitor causes an accumulation of ROS at the cell level. According to subsequent studies, these ROS cause harmful effects on non-carcinogenic cells and therefore affect the functioning of organs? Also xCT inhibitor can attack normal non-carcinogenic cells and cause damage and mutations? Thank you for discussing these hypotheses

Thank you for your comment. We do not know the harmful effects of SAS-induced ROS accumulation in non-carcinogenic cells because we have not conducted experiments in which SAS was administered to those cells. We changed “cells” to “cancer cells” in lines 287 and 290.

Reviewer 2 Report

The present investigation is about “Mechanism of cell death by combined treatment of xCT inhibitor and paclitaxel in Ovarian clear cell carcinoma”. This research provides interesting information. However, it is necessary to make some important changes before its final publication.

RESULT

General comments: I recommend not discussing or making statements about the results, this is done in the "DISCUSSION" section.

Line 65.- define “CDDP,” because it is defined up to line 83 “cytotoxicity of cisplatin”

Line 73.- Fig 1B, the literals and labels of the figures are too small, I recommend increasing them.

Line 89.- mention “not observed in in HAC-2”. remove "in".

Line 91.- They mention “Therefore, we future explored the mechanisms of cell death induced by the combined treatment of PTX and SAS.” I recommend changing this sentence to the "DISCUSSION" section; as a possible implication.

Line 113-114.- mention “Moreover, we hypothesized that the mechanism of cell death may differ depending on the levels of ROS induced by combined treatment of PTX and SAS”. I recommend moving to the "DISCUSSION" section; and discuss it.

Line 131.- mention “SAS induces ferroptosis in glioma cells [17]”. I recommend that this phrase should go in the "DISCUSSION" section.

Line 167-168.- mention “The levels of cystathionine gamma-lyase (CGL), which synthesizes cysteine form methionine via the trans-sulfuration pathway [24], were higher in HAC-2 than that in the other cell lines. The high levels of CGL suggested that combination of PTX and SAS did not induce ROS generation or cell death in HAC-2 cells”. I recommend that this sentence should be changed in the “DISCUSSION” section.

Line 173.- review this word "in-creased"

Line 177-179- mention “Combined treatment of PTX and SAS induced ferroptosis in ES-2 cells only. The key molecules of ferroptosis include GPX4, which effectively suppress lipid peroxidation [21,22]. ” I recommend that this sentence should be changed in the “DISCUSSION” section.

Line 179.- mention “The levels of GPx4 were low in ES-2 (Figure 6A).” Review this part because “fig 6A” corresponds to “A) Time course and the volume of tumors formed by RMG-1cells in BALB/cAJcl-nu/nu mice treated with PTX (20 mg/kg) and/or SAS (200 mg/kg)”. I consider that it adjusts more to "fig 5A".

Line 184.- mention “Ferroptosis is dependent on intracellular iron [19].” I recommend that this sentence should be changed in the “DISCUSSION” section.

Line 193.- mention “The lipid ROS plays an important role to induce ferroptosis [21, 22],” I recommend that this sentence should be changed in the “DISCUSSION” section.

Line 240-141.- They mention “This section may be divided by subheadings. It should provide a concise and precise description of the experimental results, their interpretation, as well as the experimental conclusions that can be drawn.” Delete this paragraph.

Line 245-256.- “Figure 6”. I recommend justifying text in this section.

DISCUSSION

In general, I recommend restructuring this section and guiding the discussion according to how the results were reported. In addition, he recommended focusing on trying to explain what were the implications or factors that caused the response. That is, the results are mentioned, but little is discussed with other investigations or an attempt is made to explain their cause.

Line 258-286.- This paragraph seems to me to be a summary of the results already mentioned above.

Line 264-265.- mention “CGL, which synthesizes cysteine form methionine via the trans-sulfuration pathway”. This was already mentioned in line 168. And it is repeated in line 272. I recommend trying to be more specific.

Line 314-321.- It is mentioned “Considering the mechanism of action of each drug, we predicted that SAS might enhance the efficacy of CDDP rather than PTX. However, SAS enhanced the CDDP cytotoxicity in only one of four OCCC cell lines and enhanced the cytotoxicity of PTX in three of four cell lines. Although the effect of SAS on tubulin and cell cycle was assessed considering the mechanism of cytotoxicity in PTX, it remains unclear whether SAS is more cytotoxic in combination with PTX than CDDP in OCCC cell lines. The effects of combination of SAS and CDDP or PTX showed different effects in different cells which could be due to the effects of CDDP or PTX.” Why is this happening?

MATERIAL AND METHODS

General comments: I recommend mentioning the criteria used to select the "n" of cells and animals used in this experimental design.

Line 464-470.- I recommend restructuring this section and specifying the handling of the data, what types of analysis were performed on each variable.

Author Response

Reviewer 2

The present investigation is about “Mechanism of cell death by combined treatment of xCT inhibitor and paclitaxel in Ovarian clear cell carcinoma”. This research provides interesting information. However, it is necessary to make some important changes before its final publication.

Thank you for your thoughtful review of our manuscript. Please find our detailed comments regarding your suggestions below.

RESULT

General comments: I recommend not discussing or making statements about the results, this is done in the "DISCUSSION" section.

Thank you for your suggestion, we deleted discussing and making statements about the results.

Line 65.- define “CDDP,” because it is defined up to line 83 “cytotoxicity of cisplatin”

As you suggested, we corrected it.

Line 73.- Fig 1B, the literals and labels of the figures are too small, I recommend increasing them.

We have done it.

Line 89.- mention “not observed in in HAC-2”. remove "in".

Thank you, we removed it.

Line 91.- They mention “Therefore, we future explored the mechanisms of cell death induced by the combined treatment of PTX and SAS.” I recommend changing this sentence to the "DISCUSSION" section; as a possible implication.

As you suggested, we replaced this sentence to the Discussion section in lines 285-286.

Line 113-114.- mention “Moreover, we hypothesized that the mechanism of cell death may differ depending on the levels of ROS induced by combined treatment of PTX and SAS”. I recommend moving to the "DISCUSSION" section; and discuss it.

As you suggested, we replaced this sentence to the Discussion section in lines 290-292.

Line 131.- mention “SAS induces ferroptosis in glioma cells [17]”. I recommend that this phrase should go in the "DISCUSSION" section.

We deleted it and descried it in lines 286.

Line 167-168.- mention “The levels of cystathionine gamma-lyase (CGL), which synthesizes cysteine form methionine via the trans-sulfuration pathway [24], were higher in HAC-2 than that in the other cell lines. The high levels of CGL suggested that combination of PTX and SAS did not induce ROS generation or cell death in HAC-2 cells”. I recommend that this sentence should be changed in the “DISCUSSION” section.

As you suggested, we changed this sentence in the Discussion section in lines 268-272.

Line 173.- review this word "in-creased"

We corrected it.

Line 177-179- mention “Combined treatment of PTX and SAS induced ferroptosis in ES-2 cells only. The key molecules of ferroptosis include GPX4, which effectively suppress lipid peroxidation [21,22]. ” I recommend that this sentence should be changed in the “DISCUSSION” section.

As you suggested, we changed this sentence in the Discussion section in lines 295-296.

Line 179.- mention “The levels of GPx4 were low in ES-2 (Figure 6A).” Review this part because “fig 6A” corresponds to “A) Time course and the volume of tumors formed by RMG-1cells in BALB/cAJcl-nu/nu mice treated with PTX (20 mg/kg) and/or SAS (200 mg/kg)”. I consider that it adjusts more to "fig 5A".

Thank you for your suggestion. We corrected it.

Line 184.- mention “Ferroptosis is dependent on intracellular iron [19].” I recommend that this sentence should be changed in the “DISCUSSION” section.

As you suggested, we changed this sentence in the Discussion section in lines 296-297.

Line 193.- mention “The lipid ROS plays an important role to induce ferroptosis [21, 22],” I recommend that this sentence should be changed in the “DISCUSSION” section.

As you suggested, we changed this sentence in the Discussion section in lines 311-312.

Line 240-141.- They mention “This section may be divided by subheadings. It should provide a concise and precise description of the experimental results, their interpretation, as well as the experimental conclusions that can be drawn.” Delete this paragraph.

We deleted this paragraph.

Line 245-256.- “Figure 6”. I recommend justifying text in this section.

As you suggested, we corrected the text in lines 231-241.

DISCUSSION

In general, I recommend restructuring this section and guiding the discussion according to how the results were reported. In addition, he recommended focusing on trying to explain what were the implications or factors that caused the response. That is, the results are mentioned, but little is discussed with other investigations or an attempt is made to explain their cause.

Line 258-286.- This paragraph seems to me to be a summary of the results already mentioned above.

As you suggested, we deleted this paragraph.

Line 264-265.- mention “CGL, which synthesizes cysteine form methionine via the trans-sulfuration pathway”. This was already mentioned in line 168. And it is repeated in line 272. I recommend trying to be more specific.

We described it only in the Discussion section in lines 268-270.

Line 314-321.- It is mentioned “Considering the mechanism of action of each drug, we predicted that SAS might enhance the efficacy of CDDP rather than PTX. However, SAS enhanced the CDDP cytotoxicity in only one of four OCCC cell lines and enhanced the cytotoxicity of PTX in three of four cell lines. Although the effect of SAS on tubulin and cell cycle was assessed considering the mechanism of cytotoxicity in PTX, it remains unclear whether SAS is more cytotoxic in combination with PTX than CDDP in OCCC cell lines. The effects of combination of SAS and CDDP or PTX showed different effects in different cells which could be due to the effects of CDDP or PTX.” Why is this happening?

We agree with you and deleted the sentence in lines 258-264.

MATERIAL AND METHODS

General comments: I recommend mentioning the criteria used to select the "n" of cells and animals used in this experimental design.

As you suggested, we described it in the Material and Methods section.

Line 464-470.- I recommend restructuring this section and specifying the handling of the data, what types of analysis were performed on each variable.

Thank you for suggestion. We do not see any particular problem based on the statements in our previous reports(Sugiyama A and Ohta T, et al. Oncology letter 2020).
